# Interdisciplinary Therapy Improves the Mediators of Inflammation and Cardiovascular Risk in Adolescents with Obesity

**DOI:** 10.3390/ijerph20237114

**Published:** 2023-11-27

**Authors:** Deborah Cristina Landi Masquio, Raquel Munhoz da Silveira Campos, Bárbara Dal Molin Netto, Joana Pereira de Carvalho-Ferreira, Carlos Roberto Bueno, Stella Alouan, Gabriela Tronca Poletto, Aline de Piano Ganen, Sergio Tufik, Marco Túlio de Mello, Nelson Nardo, Ana R. Dâmaso

**Affiliations:** 1Programa de Pós-Graduação em Nutrição, Departamento de Fisiologia, Universidade Federal de São Paulo (UNIFESP), Campus São Paulo, São Paulo 04023-061, SP, Brazil; ana.damaso@unifesp.br; 2Programa de Mestrado Profissional em Nutrição: do Nascimento à Adolescência, Curso de Nutrição, Centro Universitário São Camilo (CUSC), São Paulo 05025-010, SP, Brazil; mestradonutricao@saocamilo-sp.br; 3Grupo de Estudos da Obesidade (GEO), Universidade Federal de São Paulo (UNIFESP), São Paulo 04023-061, SP, Brazil; stella1855@gmail.com (S.A.); gabitp82@yahoo.com.br (G.T.P.); 4Programa de Pós-Graduação Interdisciplinar em Ciências da Saúde, Departamento de Biociências, Universidade Federal de São Paulo (UNIFESP), Campus Baixada Santista, Santos 11010-150, SP, Brazil; raquel.munhoz@unifesp.br; 5Programa de Pós-Graduação em Alimentação e Nutrição, Departamento de Nutrição, Universidade Federal do Paraná (UFPR), Curitiba 80210-170, PR, Brazil; barbaradm@ufpr.br; 6Laboratório Multidisciplinar em Alimentos e Saúde, Faculdade de Ciências Aplicadas, Universidade Estadual de Campinas (UNICAMP), Limeira 13484-350, SP, Brazil; joanacf@unicamp.br; 7Escola de Educação Física e Esporte de Ribeirão Preto (EEFERP), Universidade de São Paulo (USP), Ribeirão Preto 14040-900, SP, Brazil; buenojr@usp.br; 8Departamento de Psicobiologia, Universidade Federal de São Paulo (UNIFESP), São Paulo 04724-000, SP, Brazil; sergio.tufik@unifesp.br; 9Escola de Educação Física, Universidade Federal de Minas Gerais (UFMG), Belo Horizonte 31310-250, MG, Brazil; tmello@demello.net.br; 10Departamento de Educação Física, Universidade Estadual de Maringá (UEM), Maringá 87020-900, PR, Brazil; nnjunior@uem.br

**Keywords:** obesity, inflammation, leptin, adiponectin, insulin resistance, cardiometabolic risk, adolescent

## Abstract

Obesity is associated with inflammation and an increased risk of cardiovascular disease and premature mortality, as well as a range of other conditions. Obesity is a growing global problem, not only in adults, but also in children and adolescents. Therefore, the present study aimed to assess the effects of a one-year interdisciplinary intervention on the cardiometabolic and inflammatory profiles of adolescents with obesity. Twenty-two adolescents completed the intervention, which included clinical, nutritional, psychological and physical exercise counselling. Body composition, and metabolic, inflammatory, and cardiovascular risk biomarkers were analyzed before and after one year of intervention. Visceral and subcutaneous fat were determined ultrasonographically. The homeostasis model assessment of insulin resistance (HOMA-IR) and the quantitative insulin sensitivity check index (QUICKI) equation were used to estimate insulin resistance and insulin sensitivity, respectively. A reduction in body mass, adiposity, glucose, and insulin and an improved lipid profile were observed after the therapy. Hyperleptinemia was reduced from 77.3% to 36.4%. Plasminogen activator inhibitor-1 (PAI-1), intercellular adhesion molecule 1 (ICAM-1), leptin, the leptin/adiponectin ratio, and the adiponectin/leptin ratio were also significantly improved. Metabolic changes were associated with a reduction in visceral fat and waist circumference, and adiponectin and the leptin/adiponectin ratio were associated with HOMA-IR. The interdisciplinary therapy promoted improvements in hyperleptinemia and metabolic, inflammatory, and cardiovascular biomarkers.

## 1. Introduction

Obesity is a chronic disease and is related to an increase in adiposity and cardiovascular risks. In children and adolescents, the global prevalence of obesity has increased remarkably over the last years [1]. This is of particular concern, as it has been demonstrated that around 80% of adolescents with obesity will maintain the condition into adulthood [2].

In Brazil, studies undertaken before the COVID-19 pandemic estimated that the prevalence of overweight was 29.3% in children aged 5–9 years and 19.4% in adolescents [3,4]. However, the pandemic promoted changes in lifestyle habits, such as increases in sedentarism and in the consumption of ultraprocessed/fast food and sugar-sweetened beverages. Together, these factors may have resulted in an increase in the global prevalence of childhood obesity [5].

Being overweight during the pediatric phase of life can increase the risk of several comorbidities, such as dyslipidemia, type 2 diabetes, non-alcoholic fatty liver disease, metabolic syndrome, and psychological alterations [6]. Increased adipose tissue significantly influences adipocyte biology and can impair whole-body homeostasis. An excessive amount of abdominal fat, especially in the visceral depot, promotes an inflammatory state, which is characterized by the increased secretion of proinflammatory and reduced secretion of anti-inflammatory adipokines. The dysregulated secretion of pro- and anti-inflammatory adipokines results in adipose tissue dysfunction, hyperleptinemia, a proinflammatory state, an increased cardiovascular risk, and metabolic alterations [7].

Leptin is one of the most studied adipokines secreted by adipose tissue and exerts an adipostatic effect on the hypothalamus by stimulating satiety and increasing energy expenditure. However, leptin is usually found in a higher concentration in individuals with obesity, a state described as hyperleptinemia. Long-term exposure to elevated leptin levels may contribute to hypothalamic resistance, disrupted energy balance regulation, and behavioral and metabolic disorders, since it can trigger a proinflammatory state [8]. On the other hand, adiponectin is an adipokine that presents anti-inflammatory and antidiabetic properties. It is involved in a variety of physiological functions, including lipid metabolism, energy balance regulation, immune response, inflammation, and insulin sensitivity. However, adiponectin is usually found in a lower concentration in individuals with obesity [9].

Obesity is a multifactorial disease that is influenced by a variety of physiological, environmental, socioeconomic, genetic, and other factors. The consumption of excessive calories and physical inactivity are the major contributors to obesity in adolescence [10]. Therefore, the World Health Organization (WHO) has made a number of recommendations aimed at reducing obesity; these include eating more healthy food and the promotion of physical exercise and weight management [11,12].

To tackle the problem of obesity and its associated comorbidities, lifestyle interventions have been suggested, with multidisciplinary teams composed of nutritionists, exercise physiologists, psychologists, nurses, doctors, and other specialists coming together [13,14]. Previous clinical trials have demonstrated positive health outcomes in respect to intensive group interventions, which occurred 3 days a week, in terms of reducing adiposity, metabolic changes, and cardiovascular risks in children and adolescents [15,16,17]. However, given the increasing prevalence of obesity in childhood and adolescence, which has been described as an obesity epidemic by the WHO [11], there is a growing need for new, low-cost intervention models. This study, therefore, aimed to investigate the effects of a one-year interdisciplinary therapy program on the cardiometabolic and inflammatory profiles of a group of adolescents with obesity.

## 2. Materials and Methods

### 2.1. Study Design and Participants

This is a longitudinal study that investigated the effects of a one-year interdisciplinary therapy program on adolescents with obesity in the 15 to 19 age range (15.7 ± 1.2 years). The adolescents were recruited by disseminating details of the program through traditional media such as TV, radio, and newspapers in the city of São Paulo, Brazil. This study took place at the Center for Studies in Psychobiology and Exercise (CEPE) with the Obesity Study Group (GEO) of the Universidade Federal de São Paulo, in São Paulo, Brazil. The inclusion criteria for the study were to be in post-pubertal Tanner Stage V [18] and have a body mass index (BMI) > 95th percentile [19]. The exclusion criteria were the presence of a genetic disease, chronic alcohol consumption, and previous or current drug use. No financial or other incentive was offered to the subjects for participating in the therapy.

A total of 29 adolescents were recruited; however, only 22 completed the intervention with more than 75% frequency after one year. The main reasons for dropping out (*n* = 7,) were financial and family problems, school, and job opportunities. This study was conducted according to the principles laid down in the Declaration of Helsinki, was approved by the Institutional Ethical Committee (1.199.534), and was registered with ClinicalTrials.gov (NCT01358773). Informed consent or assent was obtained from all participants and/or their parents.

### 2.2. Anthropometric Measurements and Body Composition

Body weight was obtained utilizing a Filizola platform scale (Filizola S/A, São Paulo, SP, Brazil; PL 180 model) with a tolerance of 180 kg and a resolution of 100 g. Stature measurements were conducted employing a stadiometer (Sanny, São Bernardo do Campo, SP, Brazil; ES 2030 model) with an accuracy of 0.1 cm. Body mass index (BMI) was obtained by dividing the body weight (in kilograms) by the squared value of stature (in meters). The waist circumference was evaluated with subjects positioned upright with a relaxed abdomen and arms adjacent to their torso. A flexible metric tape, accurate to 1 mm, was oriented horizontally, encompassing the midpoint between the inferior boundary of the terminal rib and the iliac crest [20]. Body composition was verified using air displacement plethysmography via the BOD POD system (version 1.69, Life Measurement Instruments, Concord, CA, USA) [21].

### 2.3. Visceral and Subcutaneous Adiposity

Both visceral and subcutaneous adipose tissues were quantified using abdominal ultrasonography, with a multifrequency transducer at 3.5-MHz (broad band), and administered by a single physician with specialization in imaging diagnostics. The recorded intra-examination coefficient of variance for this ultrasonographic procedure was 0.8%. The thickness of subcutaneous adipose tissue was delineated by the span between the epidermal layer and the superficial aspect of the rectus abdominal muscle, while the visceral adipose tissue was demarcated by the distance from the deep boundary of the aforementioned muscle to the anterior facade of the aorta [22].

### 2.4. Blood Pressure Evaluation

Both Systolic (SBP) and Diastolic (DBP) blood pressures were assessed on the right upper limb utilizing a mercury gravity sphygmomanometer equipped with a suitably sized cuff. Following a minimum seated resting period of 5 min, two consecutive measurements were taken, and their average was calculated [23].

### 2.5. Biochemical Analysis

Blood samples (10 mL) were collected after a 12 h overnight fast by venous puncture and were then transferred, as appropriate, to heparinized and non-heparinized vials. Plasma glucose was determined with the aid of a commercial kit and a UniCell DXI 800 spectrophotometer (Beckman Coulter, Fullerton, CA, USA), while specific insulin (without C peptide) was determined using an enzyme assay and an ADVIA^®^ 2400 Clinical Chemistry System (Siemens, São Paulo, Brazil). Analysis of triglycerides (TG), total cholesterol (TC), high-density lipoprotein-cholesterol (HDL-c), low-density lipoprotein-cholesterol (LDL-c), and very low density lipoprotein-cholesterol (VLDL-c) were determined via enzymatic colorimetric methods (CELM, Barueri, Brazil). Some aliquots were stored at −80 °C for future analysis of adipokines and cytokines. Leptin, adiponectin, tumor necrosis factor-α (TNF-α), plasminogen activator inhibitor-1 (PAI-1), C reactive protein (CRP), intercellular adhesion molecule 1 (ICAM-1), monocyte chemotactic protein 1 (MCP1), interleukin-1 (IL-1), interleukin-6 (IL-6), interleukin-15 (IL-15), and interleukin-10 (IL-10) were determined using a multiplex assay kit (EMD Millipore; HMHMAG-34K). To mitigate daily inconsistencies, all specimens were processed simultaneously. For each analysis, a minimum of 100 beads were recorded employing a Luminex MagPix System (Austin, TX, USA), which underwent prior calibration and validation checks. The determinations of values for the unspecified samples were subsequently computed using the Milliplex Analyst Software (EMD Millipore, version 3.5.5.0, Burlington, MA, USA).

We also calculated the biomarkers representing proinflammatory leptin/adiponectin (L/A ratio) and anti-inflammatory adiponectin/leptin (A/L ratio) ratios. Leptin concentrations were analyzed based on the values proposed by Gutin et al.: 24 ng/mL for male subjects and 20 ng/mL for female subjects [24].

Homeostasis model assessment of insulin resistance (HOMA-IR) was calculated to estimate insulin resistance, employing the formula [Fasting insulin concentration (µU/mL) multiplied by fasting blood glucose concentration (mmol/L) divided by 22.5] [25]. The assessment of insulin sensitivity was obtained using the quantitative insulin sensitivity check index (QUICKI) [26], described by the equation 1/[log fasting insulin (µU/mL) + log fasting glucose (mg/dL)]. We considered as cut-off values ≥ 3.16 for HOMA-IR [27] and ≤0.339 for QUICKI [28].

Metabolic changes were assessed based on the following criteria: a waist circumference exceeding the 90th percentile for age and sex; HDL concentrations of ≤50 mg/dL in females and ≤40 mg/dL in males; triglyceride (TG) levels surpassing 150 mg/dL; blood glucose concentrations exceeding 100 mg/dL; and blood pressure readings of ≥130/85 mmHg [29].

### 2.6. Interdisciplinary Therapy

The interdisciplinary therapy had a duration of one year, from January to December, and comprised weekly two-hour group sessions. The intervention included clinical, nutritional, psychological, and physical exercise counselling, which were conducted by doctors, nutritionists, psychologists, and exercise physiologists, respectively. Figure 1 below shows a schematic model of the therapy. Detailed descriptions of the themes in each area are given below.

#### 2.6.1. Nutritional Therapy

The 60 min nutritional intervention was conducted weekly in face-to-face group sessions by a nutritionist and aimed to promote healthy eating habits and behaviors. A number of themes were discussed, such as healthy eating principles, food guides, the food pyramid, fad diets, food labels, dietetics, fat-free and low-calorie foods, fast food calories and nutritional composition, nutritional choices on special occasions, healthy sandwiches, and functional foods.

An individualized diet was prescribed for each participant, and daily energy intake was planned according to the Dietary Reference Intake (DRI) recommendations for adolescents with low levels of physical activity, considering age and sex. The distribution of macronutrients was as follows: fat (25–35%), carbohydrate (45–65%), and protein (10–30%) [30]. No pharmacotherapies, supplements, or antioxidants were recommended.

#### 2.6.2. Psychological Therapy

The adolescents took part in psychological face-to-face group support sessions to help them deal with their emotions and the relationship between their feelings and food intake. The intervention lasted 60 min and was conducted by a psychologist every two weeks. The topics discussed included depression, anxiety, emotions, eating disorders, mindful eating, low self-esteem, and body-image disorders.

#### 2.6.3. Physical Exercise Counselling

Information about lifestyle changes related to physical exercise was provided by a physical educator to encourage spontaneous physical activity (walking, stair climbing, etc.). The objective was to stimulate the adolescents to exercise three times a week (180 min/week) in their home, club, gym, or parks. The counseling included a recommendation of combined physical exercise based on the guidelines of the American College of Sports Medicine [31], comprising 30 min of aerobic training plus 30 min of resistance training. Recreational team sports (soccer, handball, basketball, etc.), gymnastics, and walking performed in groups were also encouraged. The sessions took place every two weeks and lasted 60 min.

#### 2.6.4. Clinical Intervention

Once a month, the adolescents and their parents attended a consultation with an endocrinologist to address their health and clinical parameters. The medical follow-up included taking a medical history, a physical examination, and measurements of blood pressure and body weight. The adherence of the adolescents to the therapy was also checked.

### 2.7. Statistical Analysis

Statistical analyses were executed using the STATISTICA 7.0 software (StatSoft, Tulsa, OK, USA), maintaining a statistical significance threshold of *p* < 0.05. The distribution of the numeric variables was ascertained through the Kolmogorov–Smirnov test. Variables following a parametric distribution were represented as mean values accompanied by standard deviation, while non-parametric data were denoted as medians (minimum and maximum). Paired *t* test and Wilcoxon signed rank were used for comparison between baseline and final values of parametric and non-parametric variables, respectively. Both Pearson and Spearman correlation coefficients were addressed for numerical variables. Univariate regression modeling was initiated to probe the interrelation of selected variables, designating HOMA-IR, the leptin/adiponectin ratio, and the count of metabolic changes as the dependent outcome variables. The independent variables encompassed anthropometric, metabolic, and inflammatory markers.

## 3. Results

### 3.1. Effects of the Interdisciplinary Therapy

After one year, reductions in body mass (*p* < 0.001), BMI (*p* < 0.001), fat-free mass (kg) (*p* = 0.001), waist circumference (*p* = 0.025), and visceral fat (*p* = 0.009) were observed (Table 1). Significant improvements were also found in respect to glucose (*p* = 0.001), insulin (*p* = 0.001), HOMA-IR (*p* = 0.001), QUICKI (*p* = 0.000), total cholesterol (*p* = 0.000), LDL-c (0.000), and triglycerides (*p* = 0.004) (Table 1).

In relation to the inflammatory profile, reductions were observed in leptin (*p* < 0.001) and the leptin/adiponectin ratio (*p* = 0.023). There was also an increase in the anti-inflammatory adiponectin/leptin ratio (*p* = 0.045), but no significant difference was observed in relation to the levels of adiponectin. PAI- 1 (*p* < 0.001) and ICAM 1 (*p* = 0.008) were also reduced after therapy (Table 2). Prevalence of insulin resistance decreased from 81.8% to 50.0% (*p* = 0.023), and low insulin sensibility reduced from 77.3% to 50.0% (*p* = 0.04). Hyperleptinemia also reduced from 77.3% to 36.4% (*p* = 0.027) (Figure 2).

### 3.2. Correlation and Association between Variables

Changes (Δ) in waist circumference were positively correlated with changes in glucose, PAI-1, and CRP (Figure 3). The simple regression analysis demonstrated that HOMA-IR was associated with adiponectin, the leptin/adiponectin ratio, and visceral fat. The leptin/adiponectin ratio was associated with body fat, visceral fat, and insulin. We also found that visceral fat, waist circumference, and QUICKI were predictors of the number of metabolic alterations (Table 3).

## 4. Discussion

Obesity in childhood is a significant global health problem, as excessive body fat affects the current and future health of adolescents. One of the most concerning implications of this trend is the association between excessive adipose tissue and the increased risk of developing metabolic and cardiovascular disease during adolescence and adulthood, which increases premature mortality [32]. Our weekly face-to-face interdisciplinary therapy comprised nutritional, psychological, clinical, and physical exercise counselling sessions. The design of the therapy was based on the holistic needs of the adolescents, integrating factors related to lifestyle, such as eating habits, physical activity, and emotions. Such a comprehensive approach is postulated to address not only the physical but also the psychological and behavioral aspects associated with obesity [11,12].

The primary objective of our study was to investigate the inflammatory and cardiometabolic profiles of adolescents with obesity following a one-year interdisciplinary therapy. One of the most important findings was the significant improvement in leptin after the intervention. Hyperleptinemia was significantly reduced from 77.3% to 36.4%. A previous study conducted by Dâmaso et al. [17] also reported that leptin was reduced in adolescents with obesity, from 75.0% to 55.0%, after a one-year of multidisciplinary therapy; however, this intervention took place three times a week. Another study conducted with adolescents also demonstrated improvements in leptin concentrations, promoting a 23.0% reduction after 22 weeks of semi-intensive intervention in adolescents with obesity. This model of therapy comprised weekly online health education, with only six in-person psychological support meetings, and one in-person clinical examination at the baseline [33].

Leptin is a key factor in the control of both energy balance and inflammatory processes. Previous research has consistently demonstrated that a high leptin concentration in obesity is associated with increased cardiovascular risks, behavioral disorders related to food intake, impairment in weight loss, and inflammation. Therefore, a reduction in hyperleptinemia can be considered an important outcome in relation to the control of obesity and its comorbidities [34,35].

In the present study, there was no increase in adiponectin after the intervention; however, there were improvements in both the leptin/adiponectin and adiponectin/leptin ratios. The leptin/adiponectin ratio has been considered a better predictor than leptin or adiponectin alone for cardiovascular disease, since it is considered an indicator of adipose tissue and vascular dysfunction, adiposity, and metabolic risk factors; it is useful, therefore, for cardiovascular risk stratification [35,36]. In another study, children in the highest quartile of the leptin/adiponectin ratio presented significantly higher Systolic blood pressure, CRP, triglycerides, and fasting glucose and the lowest HDL-c compared with lower quartiles [37]. Interestingly, in the present study, we observed that the leptin/adiponectin ratio was positively associated with insulin resistance, as well as with body fat, visceral fat, and insulinemia.

We also observed an increase in the adiponectin/leptin ratio after the intervention. This ratio was associated with cardiometabolic risk in children and adolescents with type 1 diabetes mellitus [38]. Moreover, this biomarker correlates better with insulin resistance than adiponectin or leptin alone, and its reduction is associated with an increase in the number of metabolic changes and has thus been proposed to be an important predictive marker for metabolic syndrome [39].

Beyond leptin and adiponectin, adipose tissue produces a variety of bioactive proinflammatory substances, including TNF-α, CRP, IL-1, IL-6, IL-15, and anti-inflammatory cytokines, such as IL-10. In obesity, adipose tissue dysfunction increases the secretion of these proinflammatory cytokines, which contributes to the development of comorbidities related to excessive adiposity, particularly in the visceral depot [40]. However, no improvements in these variables were observed in the current study. Thus, the results suggest that this type of interdisciplinary therapy can contribute to the attenuation of inflammation only in relation to improvements in leptin and the leptin/adiponectin and adiponectin/leptin ratios. Another study conducted with adolescents also failed to demonstrate any reduction in TNF-α, CRP, and IL-6 after a one-year interdisciplinary intervention, conducted tree times a week [17]. However, significant decreases in serum TNF-α and CRP were observed after a 10-week lifestyle intervention conducted with 23 children and adolescents with obesity. Probably, these results were found since this protocol proposed a moderate caloric restriction, in addition to nutritional education performed weekly through individual sessions [41].

In our study, there was a reduction in the cardiovascular risk biomarkers PAI-1 and ICAM. PAI-1 is considered a major inhibitor of the fibrinolytic system, so its elevated concentration can lead to a prothrombotic state that may contribute to the development of cardiovascular disease [42]. In adolescents with diabetes, PAI-1 levels are correlated with increased glycemia, HbA1c, triglycerides, total cholesterol, and carotid intima media thickness [43]. Likewise, ICAM-1 is also involved in cardiovascular disease development. It is a cell surface glycoprotein expressed in immune, endothelial, and epithelial cells. It acts in leukocyte rolling and in adhesive interactions with the vessel wall and guides leukocytes crossing the endothelial layer, which are involved in the atherosclerosis process. It is up-regulated in response to inflammatory cytokines and is directly related to adiposity and obesity [44]. Together, a reduction in PAI- and ICAM1 can contribute to reducing the risks of cardiovascular disease, which highlights the effectiveness of our model of therapy in controlling this outcome.

Additionally, the interdisciplinary intervention promoted a reduction in waist circumference and visceral fat. These results can be considered important outcomes, since abdominal adiposity is closely related to obesity comorbidities, such as metabolic alterations and inflammation in adolescents [7,17]. Importantly, we found that the reduction in waist circumference after therapy was positively correlated with reductions in glucose, PAI-1, and CRP. In fact, visceral fat and waist circumference are significantly and positively associated with both fasting insulin and HOMA-IR in children and adolescents [45].

In the present study, the interdisciplinary therapy was effective in improving glucose, insulin, and both indicators of insulin resistance and insulin sensitivity, HOMA-IR and QUICKI, respectively. In fact, an important reduction in the prevalence of insulin resistance was observed, with a decrease of almost 30.0%. These results are very important in clinical practice, since a high prevalence of insulin resistance is observed in obesity, and it is a predictor of body fat, cardiovascular morbidity, and end-stage renal disease [46,47]. Moreover, controlling insulin resistance can be considered as a strategy to prevent cardiovascular risks, since it is involved in the physiopathology of metabolic syndrome and the development of future artery disease, including in adolescents [48]. Indeed, in the regression analysis of the present study, the predictors of insulin resistance were visceral fat, adiponectin, and the leptin/adiponectin ratio, which reflect the well-established relationship between central adiposity, inflammatory processes, and obesity comorbidities.

Previously, a randomized controlled trial conducted with adolescents with obesity and with glucose metabolism abnormalities revealed that a 6-month Healthy Lifestyle Program, delivered twice a week, was more effective than the standard clinical care in reducing plasma insulin, 2 h plasma insulin, and HOMA-IR [49]. Reinehr et al. [50] demonstrated that a BMI standard deviation (BMI-SD) reduction of 0.25–0.5 after a lifestyle intervention was related to a decrease in HOMA-IR but also in blood pressure, triglycerides, and triglyceride/HDL-c ratio in children and adolescents. Furthermore, the study demonstrated that a reduction of more than 0.5 BMI-SD led to a more pronounced improvement in the cardiometabolic profile. This program combined physical activity, nutrition education, and behavior therapy, including the individual psychological care of the child and his or her family [50].

In the current study, in relation to the lipid profile, we observed a significant reduction in total cholesterol, LDL-c, and triglycerides. Other models of multidisciplinary therapy have similarly reported enhancements in the lipid profile of children and adolescents with obesity after one year or 22 weeks of intervention and lifestyle changes [17,33,50]. However, there was a greater improvement in triglycerides and HDL-c in children and adolescents with obesity following a higher reduction in BMI-SD (>0.50) [50].

In summary, our findings suggest that the program could be effective in attenuating cardiometabolic risk by decreasing the metabolic profile, abdominal adiposity, and visceral fat. As shown in the regression analysis, the waist circumference and visceral fat were predictors of the number of metabolic alterations. A study by Qorbani et al. [51] reported that in children and adolescents, each unit increase in BMI and in waist circumference elevates the odds of metabolic syndrome from 6 to 72% and from 1 to 20%, respectively. Obesity and metabolic syndrome in adolescence have been associated with an increased risk of coronary heart disease in adulthood [52].

A range of models using lifestyle interventions and multidisciplinary therapy have shown benefits for treating adolescents with obesity. These strategies have incorporated physical exercise, nutritional education, and behavior modification, spanning both short (2 months) and long durations (22 weeks and 12 months). The significant reductions in BMI and waist circumference reported highlight the importance of the use of this type of treatment in managing childhood obesity [14,17,50,53,54]. In a systematic review, it was demonstrated that compared with no treatment or a wait-listed control group, the use of lifestyle interventions reported a positive effect on weight loss in children and adolescents. Furthermore, studies with longer intervention periods (>6 months) showed greater weight loss than shorter therapy [14].

On the other hand, shorter-duration interventions might also foster greater adherence. Kelishadi et al. [53] introduced a 2-month lifestyle modification trial for adolescents, comprising physical exercise, diet education, and behavior modification. Remarkably, 94.7% of the participants completed the trial, with 87.3% returning for a 6-month follow-up. Additionally, we observed a lower drop-out rate (24.13%) than the intensive models (three times a week), which had a drop-out rate of 36.2% after 12 months [33].

While intensive lifestyle intervention models, conducted three times weekly, effectively controlled adiposity, cardiometabolic factors, and inflammatory profiles in adolescents with obesity [17,33], our study’s weekly interdisciplinary therapy also successfully modulated insulin resistance, hyperleptinemia, body composition, inflammation, and metabolic parameters linked to cardiovascular risk after one year. Given the childhood obesity epidemic and its implications, the prevention and management of non-communicable diseases are considered a core priority. Thus, early interventions that control adiposity, metabolic profile alterations, and the low-grade inflammatory process in adolescents with obesity are important not only for the adolescent, but also for the health of the adult they become [2]. Hence, cost-effective therapeutic models, whether interdisciplinary or of a shorter duration, should be integrated into public health policies to combat childhood obesity [12].

The main strength of this study is that it produced significant positive results in respect to obesity, which could help to reduce the comorbidities associated with obesity and improve the health of the participants as adults. The main weakness of the study is the small number of participants. However, the results of the study are important, as they encourage us to continue and further develop this line of research, but larger cohort studies are required to confirm these results.

## 5. Conclusions

In conclusion, one year of weekly interdisciplinary therapy comprising clinical, nutritional, psychological, and physical exercise counselling can contribute to a reduction in insulin resistance, hyperleptinemia, and cardiometabolic risk, improve lipid profiles, and play an important role in obesity management and reducing comorbidities in adolescents. The results observed following the intervention support its application as a model for a group approach that can be used in public health settings to support adolescents diagnosed with obesity and to improve their health and quality of life.

## Figures and Tables

**Figure 1 ijerph-20-07114-f001:**
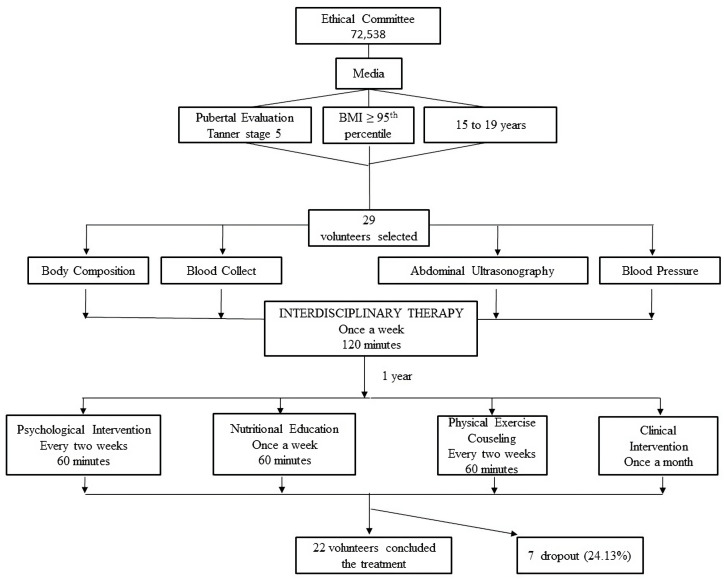
Schematic model of the interdisciplinary therapy.

**Figure 2 ijerph-20-07114-f002:**
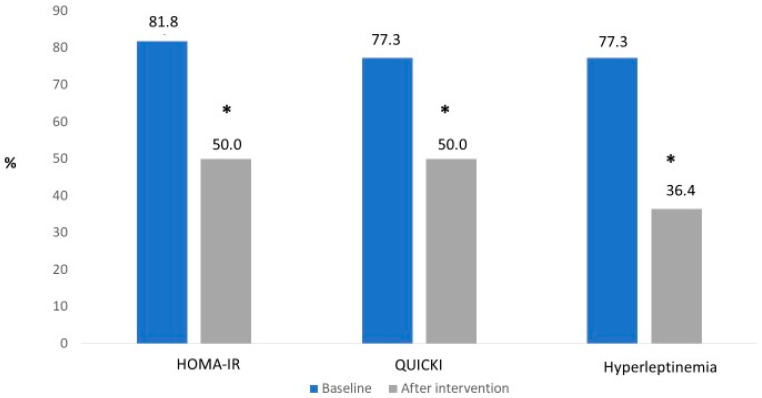
Changes in HOMA-IR, QUICKI, and hyperleptinemia between baseline and end of interdisciplinary therapy (12 months) in adolescents with obesity. * Significant difference (*p* < 0.05) between baseline and after intervention.

**Figure 3 ijerph-20-07114-f003:**
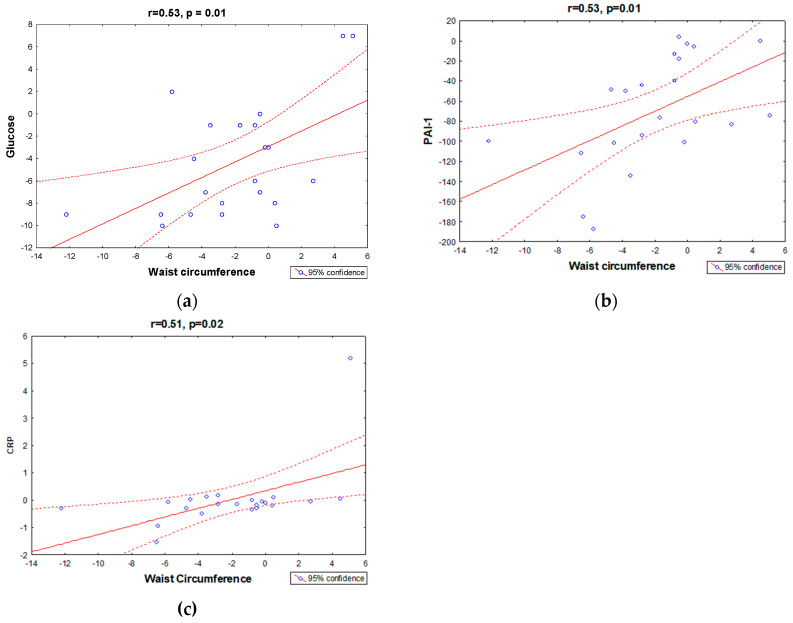
Correlation between variables: (**a**) changes (Δ) in waist circumference and glucose; (**b**) changes (Δ) in waist circumference and PAI-1; (**c**) changes (Δ) in waist circumference and CRP. Changes (Δ) in variables were calculated as: Δ = values after therapy—values at baseline.

**Table 1 ijerph-20-07114-t001:** Anthropometric, body composition, and metabolic profile of adolescents with obesity at baseline and after interdisciplinary therapy.

	Baseline	After Therapy	*p*	Δ
Age (years)	15.7 ± 1.2	16.6 ± 1.2		
Body Mass (kg)	92.6 ± 15.9	88.9 ± 16.1	<0.001	−3.7 ± 3.9
Height (m)	1.6 ± 0.1	1.6 ± 0.1	0.01	−0.0 ± 0.0
BMI (kg/m^2^)	34.5 ± 4.8	33.4 ± 4.9	0.01	−1.1 ± 1.4
Body Fat (%)	43.1 ± 5.1	42.7 ± 6.5	0.62	−0.4 ± 3.4
Fat-Free Mass (%)	56.9 ± 5.2	57.3 ± 6.5	0.62	0.4 ± 3.4
Body Fat (kg)	40.3 ± 10.3	38.4 ± 11.4	0.06	−1.8 ± 4.4
Fat-Free Mass (kg)	52.3 ± 7.7	50.3 ± 7.7	<0.001	−1.9 ±2.5
Waist Circumference (cm)	94.2 ± 11.6	92.2 ± 12.3	0.03	−2.0 ± 3.9
Visceral Fat (cm)	4.4 ± 1.1	3.9 ± 1.3	0.01	−0.5 ± 0.8
Subcutaneous Fat (cm)	3.8 ± 0.8	3.8 ± 0.9	0.89	0.0 ± 0.5
Glucose (mg/dL)	95.7 ± 5.1	91.4 ± 6.6	<0.001	−4.3 ± 5.1
Insulin (uU/mL)	15.5 ± 7.2	10.9 ± 5.8	<0.001	−4.6 ± 5.6
HOMA-IR	3.6 ± 1.7	2.5 ± 1.4	<0.001	−1.2 ± 1.3
QUICKI	0,32 ± 0,02	0,34 ± 0,03	<0.001	0.02 ± 0.02
Total Cholesterol (mg/dL)	167.8 ± 26.7	143.8 ± 26.7	<0.001	−24.0 ± 14.6
HDL-Cholesterol (mg/dL)	46.7 ± 10.2	46.6 ± 11.0	0.95	−0.09 ± 6.3
LDL-Cholesterol (mg/dL)	101.7 ± 20.3	82.8 ± 18.8	<0.001	−18.9 ± 12.4
VLDL-Cholesterol (mg/dL)	27.1 ± 36.9	14.4 ± 6.4	0.12	−12.7 ± 36.7
Triglycerides (mg/dL)	96.6 ± 48.2	72.3 ± 32.1	<0.001	−23.4 ± 35.8
Free Fatty Acids	0.7 ± 0.2	0.8 ± 0.2	0.30	0.06 ± 0.4
Systolic Blood Pressure (mmHg)	120.9 ± 8.3	118.0 ± 7.9	0.17	−3.1 ± 9.3
Diastolic Blood Pressure (mmHg)	78.1 ± 5.1	75.9 ± 4.1	0.09	−2.4 ± 5.6

Abbreviations: BMI: body mass index; HOMA-IR: homeostasis model assessment of insulin resistance; QUICKI: quantitative insulin sensitivity check index; HDL: high-density lipoprotein-cholesterol, LDL: low-density lipoprotein-cholesterol; VLDL: very low density lipoprotein-cholesterol.

**Table 2 ijerph-20-07114-t002:** Inflammatory and cardiovascular biomarkers in adolescents with obesity at baseline and after interdisciplinary therapy.

	Baseline	After Therapy	*p*	Δ
Adiponectin (ng/mL)	25.8 (9.5–97.2)	28.2 (9.5–143.3)	0.49	4.3 (−38.9–108.6)
Leptin (ng/mL)	30.8 ± 12.0	20.9 ± 14.3	<0.001	−9.9 ± 10.7
Leptin/Adiponectin ratio	1.4 ± 1.3	1.0 ± 1.6	0.02	−0.4 ± 0.8
Adiponectin/Leptin ratio	0.9 (0.2–4.6)	1.7 (0.2–17.6)	0.05	0.4 (−2.7–16.9)
TNF-α (ng/mL)	8.4 (2.3–116.0)	9.1 (1.9–48.4)	0.29	−0.7 (−67.6–5.7)
IL-1RA (pg/mL)	12.6 (0.3–327.0)	22.4 (0.3–60.1)	0.25	−0.6 (−210.7–53.4)
IL-6 (pg/mL)	1.2 (0.1–59.8)	0.9 (0.1–59.6)	0.66	−0.1 (−29.2–16.3)
IL-10 (pg/mL)	0.4 (0.1–20.3)	0.9 (0.1–30.6)	1.00	−0.1 (−3.6–12.2)
IL-15 (pg/mL)	0.1 (0.1–106.0)	0.5 (0.1–107.0)	0.30	106.0 (0.44–153.0)
CRP (ng/mL)	0.5 (0.1–1.8)	0.3 (0.1–6.3)	0.96	−0.1 (−1.5–5.2)
MCP 1 (pg/mL)	214.2 ± 106.1	168.2 ± 81.4	0.06	−23.6 ± 124.6
PAI-1 (ng/mL)	181.5 ± 61.8	111.3 ± 33.8	<0.001	−70.2 ± 54.3
ICAM (ng/mL)	146.8 ± 52.0	113.9 ± 45.6	0.01	−32.8 ± 52.9

Abbreviation: TNF-α: tumor necrosis factor-α; IL-1: interleukin-1; IL-6: interleukin-6; IL-10: interleukin-10; IL-15: interleukin-15; CRP: C reactive protein; MCP1: monocyte chemotactic protein 1; PAI-1: plasminogen activator inhibitor-1; ICAM: intercellular adhesion molecule.

**Table 3 ijerph-20-07114-t003:** Simple regression analysis.

HOMA-IR				
			−95.00%	+95.00%
	Beta (ß)	*p*	Cnf.Lmt	Cnf.Lmt
Adiponectin	−0.57	0.01	−0.08	−0.02
Leptin/adiponectin ratio	0.46	0.03	0.06	1.10
Visceral fat	0.47	0.03	0.09	1.32
Leptin/Adiponectin Ratio				
			−95.00%	+95.00%
	Beta (ß)	*p*	Cnf.Lmt	Cnf.Lmt
Body fat	0.43	0.04	0.00	0.11
Visceral fat	0.46	0.03	0.05	1.04
Insulin	0.49	0.02	0.02	0.16
Number of Metabolic Alterations				
			−95.00%	+95.00%
	Beta (ß)	*p*	Cnf.Lmt	Cnf.Lmt
Waist circumference	0.42	0.04	0.00	0.06
Visceral fat	0.45	0.04	0.02	0.68
QUICKI	0.43	0.04	−29.76	−0.25

Abbreviation: Cnf.Lmt, confidence limit.

## Data Availability

The data presented in this study are available on request from the corresponding author.

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
