# Peer review of "Interdisciplinary Therapy Improves the Mediators of Inflammation and Cardiovascular Risk in Adolescents with Obesity"

_ijerph, 2023, doi:10.3390/ijerph20237114_

Round 1

Reviewer 1 Report

Comments and Suggestions for Authors

Extensive English language editing needs to be completed throughout the entire manuscript. The manuscript is very unreadable because of so many poor usages of the English language. Some examples are below:

-Line 45: Add "the" before COVID-19 and "adverse" before changes.

-Line 49: Re-phase to "Being overweight during the pediatric phase of life can increase the risk of several comorbidities..."

-Line 94: Put space between "180" and "kg". Please do this for all quantitative values and their units.

-Line 96: Body Mass Index (BMI) "was" calculated. You have "were"

-Line 109. Why are there so many spaces before "Subcutaneous"?

-Line 180: Put "a" before endocrinologist. 

-Figure 2: You have "Insulin sensibility" as a variable. That is incredibly ignorant.

How were blood samples collected? More details are needed asides that they were collected after a 12-hour overnight fast.

Where did this interdisciplinary therapy take place? Details of the settings are needed.

Any incentives for subjects to participate in the study and remain throughout the entire duration? It would be nice to know if this was free or if they were financially motivated to participate in this study.

In the Discussion, you only cite several intervention models that have successfully improved the health of overweight/obese adolescents. There are a large number of published studies that evaluate the effectiveness of multidisciplinary interventions to treat overweight/obese adolescents. You should explore the literature more to compare your findings to what's already been done.

Your Strengths and Limitations do not provide much context nor make much sense in affecting how the audience will interpret your study and its findings.

Overall, this manuscript requires extensive editing. A significant amount of detail is further needed about the interdisciplinary therapy in the study. The Discussion should further address why this approach was effective in improving inflammation and cardiovascular risk mediators. 

Comments on the Quality of English Language

Please find a suitable co-author who is proficient in English and using it in scientific writing. They should re-evaluate the entire manuscript for you.

Author Response

REVIEWER 1

COMMENT - Extensive English language editing needs to be completed throughout the entire manuscript. The manuscript is very unreadable because of so many poor usages of the English language. Some examples are below:

-Line 45: Add "the" before COVID-19 and "adverse" before changes.

-Line 49: Re-phase to "Being overweight during the pediatric phase of life can increase the risk of several comorbidities..."

-Line 94: Put space between "180" and "kg". Please do this for all quantitative values and their units.

-Line 96: Body Mass Index (BMI) "was" calculated. You have "were"

-Line 109. Why are there so many spaces before "Subcutaneous"?

-Line 180: Put "a" before endocrinologist. 

RESPONSE - We thank the Reviewer for these comments in respect of the English language. We have made all the suggested alterations. In addition, we have thoroughly checked the English throughout the text with the help of a native speaker.

COMMENT - Figure 2: You have "Insulin sensibility" as a variable. That is incredibly ignorant.

RESPONSE - We have made appropriate changes in the names of variables.

  • Insulin resistance is now labelled HOMA-IR.
  • Insulin sensitivity is now labelled QUICKI;

COMMENT - How were blood samples collected? More details are needed asides that they were collected after a 12-hour overnight fast.

RESPONSE Thank you for this important observation. We have included more detailed information about blood sample collection.

Lines 141-150: “Blood samples (10mL) were collected after a 12-hour overnight fast by venous puncture and were then transferred, as appropriate, to heparinized and non-heparinized vials. Plasma glucose was determined with the aid of a commercial kit and a UniCell DXI 800 spectrophotometer (Beckman Coulter, Fullerton, CA, USA), while specific insulin (without C peptide) was determined using an enzyme assay and an ADVIA® 2400 Clinical Chemistry System (Siemens, São Paulo, Brazil). Analysis of triglycerides (TG), total cholesterol (TC), high density lipoprotein-cholesterol (HDL-c), low density lipoprotein-cholesterol (LDL-c) and very low-density lipopro-tein-cholesterol (VLDL-c) were determined by enzymatic colorimetric methods (CELM, Barueri, Brazil). Some aliquots were stored at -80ºC for future analysis of adipokines and cytokines”

COMMENT - Where did this interdisciplinary therapy take place? Details of the settings are needed.

RESPONSE - The interdisciplinary therapy was developed at the Center for Studies in Psychobiology and Exercise, with the Obesity Study Group (GEO) of the Universidade Federal de São Paulo, in São Paulo, Brazil. We have included this information in the manuscript.

Lines 175-184: “The interdisciplinary therapy had a duration of one-year, from January to December, and comprised weekly two-hour group sessions that took place at the Center for Studies in Psychobiology and Exercise (CEPE) with the Obesity Study Group (GEO) of the Universidade Federal de São Paulo, in São Paulo, Brazil.”

COMMENT - Any incentives for subjects to participate in the study and remain throughout the entire duration? It would be nice to know if this was free or if they were financially motivated to participate in this study.

RESPONSE - No financial or other incentive was offered to the subjects to participate in the therapy. We have included this information in the text.

Lines 106-107: “ No financial or other incentive was offered to the subjects to participate in the therapy.”

COMMENT - In the Discussion, you only cite several intervention models that have successfully improved the health of overweight/obese adolescents. There are a large number of published studies that evaluate the effectiveness of multidisciplinary interventions to treat overweight/obese adolescents. You should explore the literature more to compare your findings to what's already been done.

RESPONSE - Thank you for this observation. We have revised the Discussion to explore the literature in more detail and compare our findings with a greater number of studies.

COMMENT - Your Strengths and Limitations do not provide much context nor make much sense in affecting how the audience will interpret your study and its findings.

RESPONSE We have revised the Strengths and Limitations section to provide more context and hope this now makes more sense. Thank you for this observation.

Lines 463-468: “The main strength of this study is that it produced significant positive results in respect of obesity, which could help to reduce the comorbidities associated with obesity and improve the health of the participants as adults. The main weakness of the study is the small number of participants. However, the results of the study are important, as they encourage us to continue and further develop this line of research, but larger cohort studies are required to confirm these results.”

COMMENT - Overall, this manuscript requires extensive editing. A significant amount of detail is further needed about the interdisciplinary therapy in the study. The Discussion should further address why this approach was effective in improving inflammation and cardiovascular risk mediators. 

 RESPONSE – We have extensively edited the text, and have also now included more detail about the interdisciplinary therapy used in the study. In relation to the Discussion, we have also made significant adjustments to better address the question of why this approach was effective in improving inflammation and cardiovascular risk mediators. 

Reviewer 2 Report

Comments and Suggestions for Authors

The purpose of this study was to investigate inflammatory and cardiometabolic profile of adolescents with obesity, following one-year of interdisciplinary therapy. The authors found that leptin was significantly improved after one year intervention. Hyperleptinemia prevalence was also significantly reduced. Reduction in body mass, adiposity, glucose, insulin, and lipid profile were observed

Please add reference to the sentence at line 65-67.

The introduction needs to put more background on existing train model to manage obesity and their outcome and discuss the comparison of these models in the discussion parts regarding the model of this manuscript.

Author Response

REVIEWER 2

COMMENT - The purpose of this study was to investigate inflammatory and cardiometabolic profile of adolescents with obesity, following one-year of interdisciplinary therapy. The authors found that leptin was significantly improved after one year intervention. Hyperleptinemia prevalence was also significantly reduced. Reduction in body mass, adiposity, glucose, insulin, and lipid profile were observed

Please add reference to the sentence at line 65-67.

 RESPONSE –Thank you for this observation. We have added a reference as requested.

COMMENT - The introduction needs to put more background on existing train model to manage obesity and their outcome and discuss the comparison of these models in the discussion parts regarding the model of this manuscript.

RESPONSE – Thank you for this suggestion. We have revised the Introduction to include more details on existing models to manage obesity and their outcomes, and also improved the Discussion by comparing these models and their outcomes with those of our model.

Reviewer 3 Report

Comments and Suggestions for Authors

This study that examine the importance of using multidisciplinary therapy in management of obesity by following up 22 subjects over one year period, is interesting topic of high importance given the burden of obesity as a disease and its prevalence nowadays. However, this study over looked the comparison of the efficacy of this suggested model with other available and currently used lines of management as it might actually show the same effect!
Also, the power of the study is a big limitation as studying such effect needs more subjects than 22

Also the statistical analysis need to be described in more details in terms of describing every statistical test that was done and what other confounders that you adjusted for. 

Author Response

REVIEWER 3

COMMENT- This study that examine the importance of using multidisciplinary therapy in management of obesity by following up 22 subjects over one year period, is interesting topic of high importance given the burden of obesity as a disease and its prevalence nowadays. However, this study over looked the comparison of the efficacy of this suggested model with other available and currently used lines of management as it might actually show the same effect!

RESPONSE – We have revised the manuscript, particularly the Discussion section, in order to compare our model of treatment with other available and currently used lines of management.

- COMMENT - Also, the power of the study is a big limitation as studying such effect needs more subjects than 22.

RESPONSE –We completely agree that the number of subjects is the major limitation of our study. We have made this clear in the manuscript, and highlight the need for future studies with larger samples to confirm the results observed in our intervention model.

Lines 463-468: “The main strength of this study is that it produced significant positive results in respect of obesity, which could help to reduce the comorbidities associated with obesity and improve the health of the participants as adults. The main weakness of the study is the small number of participants. However, the results of the study are important, as they encourage us to continue and further develop this line of research, but larger cohort studies are required to confirm these results.”

- COMMENT - Also the statistical analysis needs to be described in more details in terms of describing every statistical test that was done and what other confounders that you adjusted for. 

RESPONSE - We thank the Reviewer for this pertinent comment. In response, we have included more details about the statistical analysis and the tests used. As we used a simple regression analysis, the model was not adjusted for any confounders.